# Synthesis and Antineoplastic Activity of a Dimer, Spiroindolinone Pyrrolidinecarboxamide

**DOI:** 10.3390/molecules28093912

**Published:** 2023-05-05

**Authors:** Jingyi Cui, Yujie Wang, Xiaoxin Li, Fei Xiao, Hongjun Ren, Meng Wu

**Affiliations:** 1The Key Laboratory of Geriatrics, Beijing Institute of Geriatrics, Institute of Geriatric Medicine, Chinese Academy of Medical Sciences, Beijing Hospital/National Center of Gerontology of National Health Commission, Beijing 100730, China; 2Graduate School of Peking Union Medical College, Chinese Academy of Medical Sciences, 9 DongDan Santiao, Beijing 100730, China; 3Department of Medical Research Center, State Key Laboratory of Complex Severe and Rare Diseases, Peking Union Medical College Hospital, Chinese Academy of Medical Sciences & Peking Union Medical College, Beijing 100730, China; 4Advanced Research Institute and Department of Chemistry, Taizhou University, Taizhou 318000, China

**Keywords:** MDM2 inhibitor, p53 activation, dimer spiroindolinone pyrrolidinecarboxamide, cancer treatment

## Abstract

The mutation or function loss of tumour suppressor p53 plays an important role in abnormal cell proliferation and cancer generation. Murine Double Minute 2 (MDM2) is one of the key negative regulators of p53. p53 reactivation by inhibiting MDM2–p53 interaction represents a promising therapeutic option in cancer treatment. Here, to develop more effective MDM2 inhibitors with lower off-target toxicities, we synthesized a dimer, spiroindolinone pyrrolidinecarboxamide XR-4, with potent MDM2-p53 inhibition activity. Western blotting and qRT-PCR were performed to detect the impact of XR-4 on MDM2 and p53 protein levels and p53 downstream target gene levels in different cancers. Cancer cell proliferation inhibition and clonogenic activity were also investigated via the CCK8 assay and colony formation assay. A subcutaneous 22Rv1-derived xenografts mice model was used to investigate the in vivo anti-tumour activity of XR-4. The results reveal that XR-4 can induce wild-type p53 accumulation in cancer cells, upregulate the levels of the p53 target genes *p21* and *PUMA* levels, and then inhibit cancer cell proliferation and induce cell apoptosis. XR-4 can also act as a homo-PROTAC that induces MDM2 protein degradation. Meanwhile, the in vivo study results show that XR-4 possesses potent antitumour efficacy and a favourable safety property. In summary, XR-4 is an interesting spiroindolinone pyrrolidinecarboxamide-derivative dimer with effective p53 activation activity and a cancer inhibition ability.

## 1. Introduction

As of 2020, there were 19.3 million new cancer cases and there had been nearly 10 million cancer deaths in the world, making it one of the leading causes of death for humans all over the world [1]. Changes in many oncogenes and tumour suppressor genes are associated with human cancers, these genetic changes appear at different stages from canceration to cancer cell growth, proliferation, and metastasis [2,3]. The apoptosis of normal somatic cells, which develop to a certain extent, is vital to maintain the metabolism of the human body [4]. However, the loss or mutation of tumour suppressor genes or the activation of oncogenes can lead to cell apoptosis function loss, and then induce abnormal cell proliferation and cancer generation [5,6,7]. 

Many proteins play essential roles in tumour generation and development, among which p53 is a protein encoded by the *TP53* gene that plays a crucial role in tumour suppression [8]. Under conditions of DNA damage, cellular stress, or oncogene activation situations, p53 regulates several fundamental cellular processes through transactivating target genes, such as DNA repair, cell apoptosis, and cell cycle arrest. Furthermore, these downstream reactions result in the repair or elimination of damaged and potentially tumourigenic cells [9,10,11]. Murine Double Minute 2 (MDM2) regulates the protein levels of wild-type p53 in cells at various levels, including ubiquitination mediated by ubiquitin-proteasome, the inhibition of p53 transcriptional activation by inducing p53 export to the cytoplasm, and the reduction in the degree of p53 binding to its target DNA sequence [12,13,14]. Moreover, the activation of p53 leads to its negative regulator, MDM2 overexpression, which is considered to be an autoregulatory loop [15]. The loss of p53 function induced by p53 mutation or the intracellular overexpression of MDM2 is thought to be a major cause of tumorigenesis and progression [16]. Thus, the blocking of MDM2-p53 interaction is a promising treatment approach for reactivating p53 in cancers that have wild-type or functional p53 [17].

Inspiringly, many agents have been developed based on the tactic of blocking the protein–protein interaction between p53 and MDM2, with some of them now being tested in clinical trials both in hematologic malignancy and solid tumour treatments, such as RG7388, APG-115, and ASTX-295 [18,19,20]. Nevertheless, none of these MDM2-p53 inhibitors achieved regulatory approval. One of the primary causes is due to the fact that MDM2-P53 inhibitors have several dose-related side effects that have been seen in clinical research, including the potential to induce gastrointestinal- and bone marrow-related toxicities [21,22]. Hence, it is still desirable to search for new-generation MDM2 inhibitors that have more effectiveness and tolerable toxicities.

Efforts have been made to develop novel paradigms for designing conceptually distinct MDM2-p53 inhibitors. Through the proteolysis targeting chimeras (PROTACs) strategy, some MDM2-targeted heterobifunctional PROTACs have been designed and show potent MDM2 inhibition activities [23,24]. Interestingly, with MDM2’s twin roles as an E3 ubiquitin ligase and an anticancer target, He et al. reported a series of homo-PROTACs and proved their exhilarating MDM2 self-degradation activity. These MDM2 homo-PROTACs can break down MDM2 proteins without adding additional targets and reducing the possible side effects [25]. As our previous work potently developed selective MDM2 inhibitors of PEGylation spirooxindole derivatives derived from spiroindolinone pyrrolidinecarboxamide compound **2** [26,27,28], here, we developed a dimer, spiroindolinone pyrrolidinecarboxamide XR-4, as a potent MDM2-p53 inhibitor. This compound can specifically block MDM2–p53 interaction, downregulate MDM2 protein levels, and inhibit wild-type p53 cancer cell proliferation activity in vitro. Furthermore, XR-4 has been proven to possess anti-tumour activity in vivo with favourable safety (Figure 1).

## 2. Results

### 2.1. Rational Design of Dimer Spiroindolinone Pyrrolidinecarboxamide

Based on the chemical structure of a classic MDM2 antagonist, Nutlin-3, Shu et al. developed a chiral spiroindolinone pyrrolidinecarboxamide derivative named compound **2** [28]. However, our previous work synthesized a cis-cis isomer PEGylation spirooxindole derivative, XR-2, based on compound **2** to develop a more potently selective MDM2 antagonist with acceptable toxicity [26] (Figure 1A). XR-2 presents broad-spectrum anti-tumour activity both in vitro and in vivo in various cancer types with beneficial safety properties. Inspired by He et al.’s work [25], we analysed the binding modes of compound **2** and XR-2 with MDM2 (PDB code: 5TRF [29]) using Discovery Studio 3.5 (Figure 1B). The docking results revealed that the polyethylene glycol side chain on XR-2 was more directly exposed to the solvent than its core structure, compound **2**, was; this side chain may represent a suitable position for introducing another spiroindolinone pyrrolidinecarboxamide to obtain a dimer, spiroindolinone pyrrolidinecarboxamide, which could also be regarded as a homo-PROTAC, as MDM2 belongs to the class of E3 ubiquitin ligases. Therefore, XR-4 was synthesized based on this strategy (Figure 1C).

### 2.2. Chemistry

Compound **2** was synthesized according to the literature [28]. Schema 2 shows the procedure for synthesizing critical intermediates c. A solution of 2,2′-((oxybis(ethane-2,1-diyl)) bis(oxy))diethanol and 1-chloroethyl chloroformate was stirred in CH_2_Cl_2_ at room temperature for 6 h in the presence of Et3N. Then, 200 mL of water was added, and the resulting mixture was extracted with CH_2_Cl_2_ (2 × 50 mL). The organic phase was separated and washed using brine, 1M hydrochloric acid, and water, successively. After it had been concentrated through a vacuum, the target product was obtained as an oil, which was directly used in the next step. The procedure for the synthesis of XR-4 is outlined in Figure 2. Compound **2** was dissolved in acetone; then, anhydrous Cs_2_CO_3_ and intermediate c were added, and the resulting mixture was mixed overnight at room temperature. Water was then added following the reaction, and ethyl acetate was used to extract the mixture. Separated organic phases were washed with brine, hydrochloric acid, and water, sequentially, and then concentrated under a vacuum. The target product was obtained after purification via silica gel column chromatography.

### 2.3. XR-4 Upregulates p53 Levels in Cancer Cells

Mechanistically, by binding competitively to the p53 pocket of MDM2, an MDM2 inhibitor could prevent the interaction between MDM2 and p53, resulting in p53 accumulation and translocation [17]. Herein, we first investigated whether XR-4 could promote p53 protein accumulation in different cancer cells. The 22Rv1 cell is a human castration-resistant prostate cancer cell line; this cell line is sensitive to MDM2 inhibitors as it is a p53 wild-type cell line. Western blot analysis showed that XR-4 dramatically and dose-dependently increased the amount of p53 protein after 24 h of treatment in 22Rv1 cells. Surprisingly, XR-4 exerted the activity of activating p53 at a low concentration of 0.04 μM (Figure 2A). The LNCaP cell is a human castration-sensitive prostate cancer cell line; it is also an MDM2 inhibitor-sensitive p53 wild-type cell line. Similar to the results performed in 22Rv1 cells, XR-4 also dose-dependently upregulated the p53 protein levels in LNCaP cells (Figure 2B). The HepG2 cell line is a p53 wild-type human hepatoma carcinoma cell line; thus, then chose this cell line to analyse of the p53 activation activity of XR-4. As is shown in Figure 2C, XR-4 also dose-dependently promoted p53 protein expression in HepG2 cells, and we even observed an upregulation of p53 levels under the 0.008 μM XR-4 treatment. Moreover, we then detected the time–effect relationship of XR-4 in 22Rv1 cells, and we found that under the 5 μM XR-4 treatment for 4 h, the p53 protein levels were significantly upregulated, while it seemed to have no influence on p53 protein levels under the XR-4 treatment for 2 h, and the XR-4 treatment for 8 h was thought to induce the strongest p53 activation activity (Figure 2D). Finally, we detected whether XR-4 influences the mRNA levels of p53 via conducting quantitative polymerase chain reaction (qPCR) analysis; the results indicated that XR-4 had almost no influence on p53 mRNA expression levels under our chosen concentration treatment both in LNCaP and HepG2 cells (Figure 2E,F). Together, our results proved that XR-4 could potently promote p53 protein accumulation in different cancer cells in both dose-dependent and time-dependent manners by blocking the MDM2-p53 interaction.

### 2.4. XR-4 Activates p53 Downstream Target Genes

As our previous work demonstrated that XR-4 induced p53 accumulation in cancer cells, we then performed qRT-PCR analysis to investigate the influence of XR-4 on p53 target genes in different cancer cells. The *p21* gene is one of the p53 signal pathway downstream genes, and *p21* is a crucial cell cycle regulatory gene. It is involved in the process of cell growth, differentiation, senescence, and death and is closely related to tumorigenesis [30]. The qRT-PCR results revealed that XR-4 dose-dependently increased *p21* mRNA expression levels, both in LNCaP and HepG2 cells (Figure 3A,C). The p53 downstream target gene, *PUMA,* plays an important role in mediating p53-induced cell death [31]. Similarly, our qRT-PCR results also revealed that XR-4 could dose-dependently upregulate *PUMA* mRNA expression levels both in LNCaP and HepG2 cells (Figure 3B,D). These results surely confirmed that XR-4 can activate the p53 pathway.

### 2.5. XR-4 Downregulates MDM2 Protein Levels 

Homo-PROTACs are a type of PROTAC molecule that consists of two identical E3 ligase-targeted ligands. As the structure of XR-4 could be regarded as a homo-PROTAC molecule, in consideration of its effective p53 pathway activation activity, we chose to use XR-4 to further evaluate its effects on MDM2 degradation in 22Rv1 cell lines via Western blot analysis. The results showed that XR-4 dose-dependently induced MDM2 protein cleavage in 22Rv1 cells (Figure 4A,B). The MDM2 protein DC_50_ of XR-4 was about 5.0 μM. These results preliminary proved that XR-4 could play a role as a homo-PROTAC molecule. 

### 2.6. XR-4 Suppresses the Viability of Wild-Type p53 Cancer Cells Both In Vitro and In Vivo 

As XR-4 was proven to upregulate the p53 key downstream target genes, such as *p21*, *PUMA* and *MDM2*, *p21* and *PUMA* can regulate cell cycle arrest, apoptosis, and senescence in various cancer cell lines. Subsequently, we detected the cell proliferation inhibition activities of XR-4 in different wild-type p53 cancer cell lines via the CCK-8 (Cell Counting Kit-8) assay, including LNCaP (prostate cancer), 22Rv1 (prostate cancer), HepG2 (liver cancer), HCT116 (colorectal cancer), MCF7 (breast cancer), and T24 (bladder cancer) cell lines. Moreover, we chose the MDM2 inhibitor, RG7388, which is under clinical research as a positive control. The results revealed that XR-4 showed comparable cell proliferation inhibition activities to RG7388 in all the detected wild-type p53 cancer cell lines (Table 1). Notably, we also detected the impact of XR-4 on DU145 (prostate cancer) and PC-3 (prostate cancer) cell lines, as DU145 is a p53 mutated cell line, while PC-3 is a p53 null cell line. As is shown in Table 1, the DU145 and PC-3 cell line proliferation inhibition IC_50_ values of XR-4 were both over 50 μM, while the DU145 proliferation inhibition IC_50_ of RG7388 was 12.6 μM, and the PC-3 proliferation inhibition IC_50_ of RG7388 was 21.4 μM, which are both lower than that of XR-4. These results indicate that XR-4 is an effective and selective MDM2 inhibitor with lower off-target cell toxicity than RG7388 has. 

To evaluate the influence of XR-4 on cancer cells’ clone formation ability, 22Rv1 cells were treated with different concentrations of XR-4 for about 2 weeks; the results showed that the XR-4 treatment more effectively reduced the number of 22Rv1 cell colonies in comparison to that of the negative control (Figure 5A). Western blot analysis also indicated that XR-4 could induce the accumulation of cleaved PARP, an apoptotic marker protein, in LNCaP cells in a dose-dependent manner (Figure 5B). Furthermore, flow cytometry analysis also revealed that XR-4 induced LNCaP cell apoptosis (Figure 5C). These results suggest that XR-4 could suppress cell proliferation and induce apoptosis through p53 activation in vitro.

Next, we investigated the in vivo tumour inhibition activity of XR-4 using a 22Rv1 xenograft model. As is shown in Figure 6A, 50 mg/kg of XR-4 more potently suppressed the 22Rv1 xenograft growth compared with that of the negative control group. The tumour growth inhibition rate was about 66.7% after 15 days of XR-4 treatment. Importantly, the 50 mg/kg XR-4 treatment showed almost no influence on the mice’s body weight (Figure 6B). Together, our in vivo research demonstrated that XR-4 possessed potent antitumour efficacy and favourable safety properties.

## 3. Conclusions

In summary, we synthesized a potent MDM2-p53 inhibitor, XR-4, which possesses a dimer, spiroindolinone pyrrolidinecarboxamide, chemical structure. XR-4 can selectively promote wild-type p53 accumulation in cancer cells, and then activate the downstream target genes, *p21* and *PUMA,* of the p53 pathway to inhibit cancer cell proliferation and induce cell apoptosis. XR-4 showed comparable cancer cell proliferation inhibition activity and lower off-target toxicity compared to those of RG7388 in a broad of spectrum p53 wild-type cancer cells, including liver cancer, prostate cancer, breast cancer, bladder cancer, and colorectal cancer cells. Importantly, XR-4 showed potent antitumour efficacy and desirable safety properties in vivo.

Notably, our works also preliminarily proved that XR-4 could perform as a homo-PROTAC lead compound that degrades MDM2 protein, which is worthy of a future study on the structure–activity relationship. All in all, XR-4 is an interesting spiroindolinone pyrrolidinecarboxamide-derivative dimer with effective p53 activation activity and feasible lower off-target toxicity.

## 4. Materials and Methods

### 4.1. Chemistry

Bis(1-chloroethyl) (((oxybis(ethane-2,1-diyl))bis(oxy))bis(ethane-2,1-diyl)) bis(carbonate) (linker intermediate): The solution of 2,2’-((Oxybis(ethane-2,1-diyl)) bis(oxy)) diethanol (1.94 g, 10 mmol, purchased from Sigma-Aldrich, Darmstadt, Germany) and 1-chloroethyl chloroformate (4.29 g, 30 mmol, purchased from Sigma-Aldrich) was stirred in CH_2_Cl_2_ (100 mL, purchased from Bidepharm, Shanghai, China) at room temperature for 6 h in the presence of Et_3_N (4.04 g, 52.5 mmol, purchased from Bidepharm, Shanghai, China). Then, 200 mL of water was added, and the resulting mixture was extracted with CH_2_Cl_2_ (2 × 50 mL). The organic phase was separated, and subsequently, washed with brine, 1M hydrochloric acid, and water. After being concentrated through a vacuum, the target product was obtained as an oil, which was directly used in the next step.^1^H NMR (400 MHz, CDCl_3_): 6.30 (q, *J* = 5.8 Hz, 1H), 4.29 (dd, *J* = 3.12 Hz, 2H), 3.68 (dd, *J* = 4.6 Hz, 2H), 3.59 (s, 4H), 1.76 (d, *J* = 5.8 Hz, 3H).((Oxybis(ethane-2,1-diyl))bis(oxy))bis(ethane-2,1-diyl) (2′S,2′′′S,3R,3″R,4′S,4′′′S,5′R,5′′′R)-bis(5′-((4-carbamoyl-2-methoxyphenyl)carbamoyl)-6-chloro-4′-(3-chloro-2-fluorophenyl)-2′-neopentyl-2-oxospiro[indoline-3,3′-pyrrolidine]-1′-carboxylate) (XR-4): Compound **2** was synthesized according to the literature [28]. Compound **2** (612 mg, 1.0 mmol) was dissolved in acetone (100 mL, purchased from Bidepharm, Shanghai, China), and then anhydrous Cs_2_CO_3_ (2.0 mmol, purchased from Sigma-Aldrich, Darmstadt, Germany) and bis(1-chloroethyl) (((oxybis(ethane-2,1-diyl))bis(oxy))bis(ethane-2,1-diyl)) bis(carbonate) (204 mg, 0.5 mmol) were added, and the resulting mixture was stirred overnight at room temperature. The reaction mixture was then diluted with water (20 mL) and extracted with ethyl acetate (2 × 50 mL, purchased from Bidepharm, Shanghai, China) and washed with brine, hydrochloric acid, and water, before being concentrated through a vacuum. After purification via silica gel column chromatography, the target product, XR-4, was obtained (405 mg, yield: 55%; purity: 97%) as a white solid. The melting point of XR-4 was about 155–156 °C. ^1^H NMR (CDCl3, 400 MHz): 10.59 (s, 1H), 8.44 (d, *J* = 8.24 Hz, 1H), 7.79 (s, 1H), 7.54 (s, 1H), 7.48 (t, *J* = 6.90 Hz, 1H), 7.28–7.39 (m, 2H), 7.17–7.25 (m, 2H), 7.04 (t, *J* = 7.96 Hz, 1H), 6.20–6.55 (bs, 1H), 5.35–5.65 (bs, 1H), 4.69 (t, *J* = 8.84 Hz, 1H), 4.35–4.55 (m, 3H), 3.92 (s, 3H), 3.81 (t, *J* = 4.46 Hz, 2H), 3.60–3.75 (m, 5H), 3.24 (t, *J* = 10.7 Hz, 1H), 1.78 (s, 1H), 0.85–1.00 (m, 10H).^13^C NMR (CDCl_3_, 101 MHz): 174.09, 171.38, 168.83, 157.38, 154.90, 149.29, 148.00, 139.75, 134.55, 130.00, 129.75, 128.43, 127.15, 125.26, 124.62, 124.49, 123.54, 123.10, 121.19, 119.97, 117.76, 115.80, 109.62, 70.48, 68.35, 67.66, 66.32, 66.28, 65.16, 55.39, 50.54, 42.50, 30.15, 29.61, 29.50. HRMS (ESI-TOF): *m*/*z* calculated for C_72_H_78_Cl_4_F_2_N_8_NaO_15_^+^ [M + 2H + Na]^+^: 1495.4207, found: 1495.4204. The NMR and HRMS data of XR-4 were showed in Appendix A. 

### 4.2. Cell Culture

LNCaP (ATCC, CRL-1740), 22Rv1 (ATCC, CRL-2505), PC-3 (ATCC, CRL-1435), and DU145 (ATCC, HTB-81) human prostate cancer cell lines were purchased from the American Type Culture Collection (ATCC). The human colorectal cancer cell line, HCT116 (CL-0096), human liver cancer cell line, HepG2 (CL-0103), human breast cancer cell line, MCF7 (CL-0149), and human bladder cancer cell line, T24 (CL-0227) were purchased from Procell Life Science & Technology Co., Ltd., Wuhan, China. All cell lines were cultured in a 37 °C humidified incubator with 5% CO_2_ and cultured in medium containing 10% foetal bovine serum (FBS) (Gibco, Waltham, MA, USA). 22Rv1, LNCaP, PC-3, and T24 cells were cultured in Roswell Park Memorial Institute (RPMI) 1640 medium (Meilunbio, Dalian, China). MCF7, HepG2, DU145, and HCT116 cells were cultured in Dulbecco’s Modified Eagle Medium (DMEM) (Meilunbio, Dalian, China).

### 4.3. RNA Extraction, Reverse Transcription, and Quantitative Real-Time Polymerase Chain Reaction (qPCR) 

The different types of cancer cells were seeded at a density of ~4 × 10^5^ cells per well in six-well plates. After 48 h, vehicle (0.2%DMSO) or test compounds were applied at designated concentrations. After another 24 h of incubation, 500 µL of TRIzol (Invitrogen, Carlsbad, CA, USA) was added in each well of six-well plates. A colourless upper aqueous phase was obtained by adding 100 µL of chloroform to 500 µL of TRIzol reagent. Precipitation with isopyknic isopropanol was performed on the RNA-containing aqueous phase. RNA precipitation was washed with 75% ethanol and resuspended using the appropriate volume of RNase-free water. The total RNA was reverse-transcribed using a reverse transcription kit (Accurate Biology, Hunan, China). cDNA was used for qPCR using the 2× SYBR Green Mix (Accurate Biology, Hunan, China). The calculation method of qPCR was the 2^−ΔΔCt^ method, with GAPDH as the housekeeping gene. The specific qPCR primers were as follows: p53 forward, 5′-CCTCAGCATCTTATCCGAGTGG-3′; reverse, 5′-TGGATGGTGGTACAGTCAGAGC-3′; *p21* forward, 5′-ATGAAATTCACCCCCTTTCC-3′; reverse, 5′-CCCTAGGCTGTGCTCACTTC-3′; *PUMA* forward, 5′-GACGACCTCAACGCACAGTA-3′; reverse, 5′-AGGAGTCCCATGATGAGATTGT-3′; *GAPDH* forward, 5′-GGTATCGTGGAAGGACTCATGAC-3′; reverse, 5′-ATGCCAGTGAGCTTCCCGTTCAG-3′.

### 4.4. Western Blot 

Cancer cells were seeded at a density of ~4 × 10^5^ cells per well in six-well plates. After 48 h, vehicle (0.2% DMSO) or test compounds were applied at designated concentrations. After another 24 h of incubation, cells were washed twice with phosphate-buffered saline (PBS), and protein samples were obtained by using RIPA lysis buffer (Beyotime Biotechnology, Shanghai, China). Sodium dodecyl sulphate–polyacrylamide gel electrophoresis (SDS-PAGE) was used to separate protein samples, which were then transferred to polyvinylidene fluoride (PVDF) membranes. After blocking for 1 h at room temperature with a blocking buffer (Beyotime Biotechnology, Shanghai, China) for 1 h, the membranes were incubated overnight at 4 °C with primary antibodies. The primary antibodies used in this research included rabbit anti-MDM2 (Abways, Tracxn Technologies Limited, India), rabbit anti-p53 (Abways, Tracxn Technologies Limited, India), rabbit anti-cleaved PARP1 (Santa Cruz, CA, USA), mouse anti-GAPDH (Abmart, Shanghai, China), and mouse anti-β-actin antibody (Abmart, Shanghai, China). Following three washes with TBST buffer for 10 min each, the membranes were incubated for 1 h at room temperature with the anti-mouse or anti-rabbit secondary antibodies (Abmart, Shanghai, China). After being washed with TBST solution three times, protein bands were detected via an enhanced chemiluminescence (ECL) solution and an Image Analysis System (Sinsage, Beijing, China). All the experiments were performed in triplicates.

### 4.5. Cell Viability Assay

Cell counting was performed using a Neubauer haemocytometer. Approximately 3000 cells per well were seeded in 96-well plates for 24, and then incubated with DMSO (0.5%), XR-4, or RG7388 at indicated concentrations for 72 h. Counting Kit-8 (CCK-8) solution (Meilunbio, Dalian, China) was used to measure the cell viability. CCK8 reagent was configured into a medium containing 10% CCK-8 solution and added in the form of exchange fluid. Ninety-six-well plates supplemented with CCK-8 solution were incubated at 37 °C for 4 h. Then, the 96-well plate was taken out, and the OD value of each well at a wavelength of 450 nm was measured using a microplate reader (BioTek, Bedfordshire, UK). All experiments were performed in triplicates. 

### 4.6. Colony Formation Assay 

Cell counting was performed using a Neubauer haemocytometer. About 1000 22Rv1 cells in each well were seeded in six-well plates. Cells were treated with DMSO (0.2%) or XR-4 at the indicated concentration for 14 days. The cell culture medium was changed regularly. Following washing with PBS, the colonies were fixed with methanol for 15 min. After removing the fixative, colonies were stained with 0.1% crystal violet (Solarbio, Beijing, China) for 15 min and washed with running water. After each well of the six-well plate was dried in the air, colonies were photographed with a camera.

### 4.7. Flow Cytometry Assay

The LNCaP cells were seeded at a density of about 4 × 10^5^ cells per well in a 6-well plate. After being incubated for 48 h, DMSO (0.2%) and XR-4 were added to each well at designated concentrations. After the compound treatment for 24 h, LNCaP cells were collected with EDTA-free Tyrisin (Beyotime, Beijing, China) and washed twice with cold PBS. Then, cells were incubated for 15 min at room temperature with Annexin V-FITC-PI in a binding buffer. Cell staining was performed, and staining cells were collected on a flow cytometer (BDC6, BD Biosciences, San Jose, CA, USA). Then, data were analysed using FlowJo software. Results are expressed as percentages of Annexin V+ cells.

### 4.8. In Vivo Studies

The mice used in this study, 4-6-week-old BALB/c male nude mice, were purchased from Shanghai Laboratory Animal Center (Shanghai, China). The 22Rv1 cells (5 × 10^6^ cells with Matrigel at a ratio of 1:1) were subcutaneously injected into the left flanks of the mice. When the average tumour volumes reached about 100–150 mm^3^, the mice were assigned randomly to 2 groups consisting of 8 mice each. XR-4 was administered once daily intraperitoneally for 15 days. Tumour volume and mice’s body weight were measured every other day, and data were recorded and are represented as mean ± SD. The tumour volume (V) was calculated using the formula V = (L × W^2^)/2 (mm^3^), where L is the longest diameter, and W is the diameter perpendicular to L of the tumour. All animal experiments were approved by the KeyGEN BioTECH Institutional Animal Care and Use Committee (Reference number: IACUC-20210302). 

### 4.9. In Silico Docking

The molecular docking of compound **2** and the XR-2 to MDM2 protein (PDB code: 5TRF) was conducted as described in our previous works [26]. The crystal structure of MDM2 (PDB code: 5TRF) was downloaded from the PDB database (https://www.rcsb.org, accessed on 24 October 2020). Chain A of the MDM2 crystal structure was remained and was modified by the ‘protonate 3D’ module of Discovery Studio 3.5. Two-dimensional chemical structures of compound **2** and XR-2 were drawn using the Chemdraw 18.1 software, and then were generated in three-dimensional structures using the ‘prepare ligands’ module Discovery Studio 3.5. The binding site was centred at Ile99 in MDM2 with a radius of 10 Å to cover the binding pocket of MDM2. Then, the prepared compounds were docked into the MDM2 chain A binding site using the ‘Libdock’ module of Discovery Studio 3.5 in default mode. After analysing the 10 binding poses of compound **2** and XR-2 to MDM2, we selected the highest-ranked pose for the MDM2 structure as the binding model of compound **2** and XR-2. 

### 4.10. Statistical Analysis 

Statistical significance was defined as *p*-values less than 0.05. Student’s *t*-test was used to compare the two groups. One-way ANOVA analysis of variance was performed to compare two or more groups. GraphPad Prism software (version 8.0) was used to perform statistical analysis on the three independent studies.

## Data Availability

Not applicable.

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
