# Peer review of "Synthesis and Antineoplastic Activity of a Dimer, Spiroindolinone Pyrrolidinecarboxamide"

_molecules, 2023, doi:10.3390/molecules28093912_

Round 1

Reviewer 1 Report

The manuscript entitled " Synthesis and Antineoplastic Activity of a Dimer Spiroindo- 2 linone Pyrrolidinecarboxamide" describes the synthesis of a dimer spiroindolinone pyrrolidinecarboxamide XR-4 and evaluation of inhibition on MDM2, upregulation of p53 target genes p21 and PUMA levels, and apoptosis induction of XR-4 in different cancers. This manuscript is interesting and I advise publishing it after the alterations listed below:

1.     For evaluation of XR-4 on cancer cell clone formation ability, 22Rv1 cancer cell line was selected. What was the logic? The IC50 value of XR-4 on LN-CAP was 0.043 µM and it seems this cell line could be appropriate for further assays.

 2.     Section 2.6. XR-4 suppresses wild-type p53 cancer cells viability both in vitro and in vivo: Why the cancer cells was treated with 0.2 μM XR-4?

 3.     The solvent, DMSO, was used as vehicle. The concentration should be mention.

4.     Fig.5B: The Molecular weight of cleaved PARP is not clear.

5.     PARP cleavage is not specific indicator for apoptosis. The other assays such as Flow cytometry is needed for confirmation.

6.     Figs. The molecular weight must be included in all gels.

7.     The materials and method section is not well presented.

8. The grammatical, writing and typos errors must be corrected thoroughly in the manuscript. 

Author Response

Point 1: For evaluation of XR-4 on cancer cell clone formation ability, 22Rv1 cancer cell line was selected. What was the logic? The IC50 value of XR-4 on LN-CAP was 0.043 µM and it seems this cell line could be appropriate for further assays.

Response: Thank you for this suggestion. Although the CCK8 assay results indicated that XR-4 was more sensitive to LNCaP cells, we performed cell colony formation assay by both LNCaP and 22Rv1 cells, unfortunately, the clone formation ability of LNCaP cells was quite feeble and we didn’t gain suitable pictures as the LNCaP cells hardly generation cell colony in our experiments. So, we only showed the results gained in 22Rv1 cells (Figure 5A).

Point 2: Section 2.6. XR-4 suppresses wild-type p53 cancer cells viability both in vitro and in vivo: Why the cancer cells was treated with 0.2 μM XR-4?

Response: Thank you very much for this recommendation. To fully investigate the influence of XR-4 on cancer cells colony formation activity, in the revised manuscript, we showed that XR-4 could dose-dependently inhibit 22Rv1 cells colony formation ability in different concentrations. (Figure 5A).

Point 3: The solvent, DMSO, was used as vehicle. The concentration should be mention.

Response: Thank you for this recommendation. We have added the descripotion of DMSO concentration to the methods. In addition, in the figures of the revised manuscript, we list DMSO (vehicle) to replace XR-4 0 μM.

Point 4: Fig.5B: The Molecular weight of cleaved PARP is not clear.

Response: Thank you for this suggestion. We marked the molecular weight of cleaved PARP protein in the the revised manuscript (Figure 5B).

Point 5: PARP cleavage is not specific indicator for apoptosis. The other assays such as Flow cytometry is needed for confirmation.

Response: Thank you for this suggestion. To explain the influence of XR-4 on wild type p53 cancer cells apoptosis, we performed flow cytometry assay in the revised manuscript, the results indicated that XR-4 could induce wild-type p53 cancer cell line LNCaP cells apoptosis (Figure 5C).

Point 6: Figs. The molecular weight must be included in all gels.

Response: Thank you for this suggestion. We marked the molecular weight of all the western blot results in the the revised manuscript.

Point 7: The materials and method section is not well presented.

Response: Thank you very much for this recommendation. We re-write the materials and method section in the revised manuscript to better describe our experiment methods.

Point 8: The grammatical, writing and typos errors must be corrected thoroughly in the manuscript.

Response: Thank you for the kindly suggestion. The language in the revised manuscript has been polished by MDPI official English editing (English-65827) .  

Reviewer 2 Report

The manuscript entitled “Synthesis and Antineoplastic Activity of a Dimer Spiroindolinone Pyrrolidinecarboxamide” by Jingyi Cui et al., describes a new dimeric spiroindole derivative as a potential antitumor agent. The manuscript is logically structured. However, a few remarks should be noted:

1.   Chemistry 2.2. Lines 108-117 should be removed. Add results (for instance, a table) with optimization of synthetic methodology (solvents, bases, temperature, etc).

2.     Materials and Methods. Add information about chemical suppliers, NMR and HRMS.

3. Characterization of compounds (Materials and Methods). Indicate the purity of synthesized compounds. Also, add information about melting or boiling points for the compounds.

4. Molecular docking (Materials and Methods). Add information about the protein and receptor preparation, structure optimization of compounds, docking (SP, XP), etc.

5. In vitro and in vivo studies. The positive control is missing. To better understand the significance of the obtained results, the authors should add positive control.

6. Toxicity. Add information about the toxicity of the compound to normal mammalian cells.

6. Supplementary. Add high-quality NMR and high-resolution mass spectroscopy figures to the Supplementary.

Author Response

Point 1: Chemistry 2.2. Lines 108-117 should be removed. Add results (for instance, a table) with optimization of synthetic methodology (solvents, bases, temperature, etc).

Response: Thank you for this suggestion. We are sorry to this comment, as we didn’t perform the optimization of synthetic methodology of XR-4 in this manuscript, the lead compound (compound 2) was synthesized according to the literature (Shu L, et al. Organic Process Research & Development, 2013), and XR-4 was synthesized based on our previous works (Meng W, et al. Frontiers in pharmacology, 2022). We think it’s necessary to perform the optimization of synthetic methodology in our future works. Thanks again for this insightful suggestion.

Point 2: Materials and Methods. Add information about chemical suppliers, NMR and HRMS.

Response: Thank you for this suggestion. We added the chemical suppliers in the Materials and Methods, and the NMR and HRMS data of XR-4 were showed in supplementary materials in the revised manuscript.

Point 3: Characterization of compounds (Materials and Methods). Indicate the purity of synthesized compounds. Also, add information about melting or boiling points for the compounds.

Response: Thank you for this suggestion. The purity and melting points of XR-4 were showed in Materials and Methods (line 308-309) in the revised manuscript.

Point 4: Molecular docking (Materials and Methods). Add information about the protein and receptor preparation, structure optimization of compounds, docking (SP, XP), etc.

Response: Thank you for this suggestion. We added some data of Molecular docking information in the Materials and Methods (line 413-425) in the revised manuscript.

Point 5: In vitro and in vivo studies. The positive control is missing. To better understand the significance of the obtained results, the authors should add positive control.

Response: Thank you for this kind suggestion. RG7388 (Idasanutlin) is a MDM2 inhibitor which is under clinical research, RG7388 could induce p53 protein accumulation and upregulate p53 downstream target gene levels reported in other literatures, we chose RG7388 as positive control to compare the cancer cell proliferation inhibition activity between XR-4 and RG7388 (Table 1), the results proved that XR-4 showed comparable cell proliferation inhibition activities to RG7388 in all these detected wild-type p53 cancer cell lines. However, we have not compared the in vivo activities of XR-4 and RG7388 mainly because we have some difficult to obtain abundant RG7388 to perform in vivo evaluation in our previous works.   

Point 6: Toxicity. Add information about the toxicity of the compound to normal mammalian cells.

Response: Thank you for this suggestion. We have made some efforts to investigate the toxicity of XR-4, we detected the influence of XR-4 on DU145 (prostate cancer) and PC-3 (prostate cancer) cell lines, as DU145 is a p53 mutated cell line, while PC-3 is a p53 null cell line. As is shown in table 1, the DU145 and PC-3 cell lines proliferation inhibition IC50 of XR-4 were both over 50 μM, quite higher than that in p53 wild-type cancer cell lines, while the DU145 proliferation inhibition IC50 of RG7388 was 12.6 μM, the PC-3 proliferation inhibition IC50 of RG7388 was 21.4 μM, both lower than that of XR-4. These results indicated that XR-4 may have lower off-target cell toxicity than RG7388. Besides, the in vivo assay also indicated that XR-4 50 mg/kg didn’t influence the mice body weight (Figure 6B), which also preliminarily proved the safety of XR-4.  

Point 7: Supplementary. Add high-quality NMR and high-resolution mass spectroscopy figures to the Supplementary.

Response: Thank you for this suggestion. The NMR and HRMS data of XR-4 were showed in supplementary materials in the revised manuscript.

Reviewer 3 Report

This manuscript reports the synthesis of a potent MDM2-p53 inhibitor XR-4 which possessed a PROTAC like homo-dimer of spiroindolinone pyrrolidinecarboxamide. Authors identified that XR-4 selectively promotes wild-type p53 accumulation in cancer cells, then activate the downstream target genes p21 and PUMA of the p53 pathway that inhibits cancer cell proliferation and induce apoptosis. XR-4 exhibited comparable anti-proliferative effect and lower off-target toxicity in comparison to control molecule (RG7388) in a broad spectrum p53 wild-type cancer cells. Moreover, XR-4 also exhibited potent in vivo antitumor efficacy and desired safety.  In addition, authors proved that XR-4 could perform as a homo-PROTAC compound that degrades MDM2 protein. This work is an extension of their previous report on XR-2 inhibition of MDM2 in prostate cancer. However, XR-4 seems to be an interesting dimer molecule with high p53 activation activity and lower off-target toxicity. Except some grammatical issues, this manuscript is presented well with appropriate references. I recommend this manuscript for publication in the journal of Molecules.

Author Response

Point 1: Except some grammatical issues, this manuscript is presented well with appropriate references. I recommend this manuscript for publication in the journal of Molecules.

Response: Thank you for this suggestion. The language of the manuscript has been polished by MDPI official English editing (English-65827) in the revised manuscript.

Round 2

Reviewer 1 Report

I recommend that the revised paper could be accepted.

Reviewer 2 Report

All my previous remarks were taken into account. Therefore, the manuscript can be accepted for publication in the present form.